# Blind Turbo Equalization of Short CPM Bursts for UAV-Aided Internet of Things

**DOI:** 10.3390/s22176508

**Published:** 2022-08-29

**Authors:** Zihao Pan, Chen Xie, Heng Wang, Yimin Wei, Daoxing Guo

**Affiliations:** College of Communication Engineering, Army Engineering University of PLA, Nanjing 210001, China

**Keywords:** blind equalization, continuous-phase modulation, EM algorithm, Lazy Viterbi algorithm

## Abstract

With the surge of Internet of Things (IoT) applications using unmanned aerial vehicles (UAVs), there is a huge demand for an excellent complexity/power efficiency trade-off and channel fading resistance at the physical layer. In this paper, we consider the blind equalization of short-continuous-phase-modulated (CPM) burst for UAV-aided IoT. To solve the problems of the high complexity and poor convergence of short-burst CPM blind equalization, a novel turbo blind equalization algorithm is proposed based on establishing a new expectation–maximization Viterbi (EMV) algorithm and turbo scheme. Firstly, a low complexity blind equalization algorithm is obtained by applying the soft-output Lazy Viterbi algorithm within the EM algorithm iteration. Furthermore, a set of initializers that achieves a high global convergence probability is designed by the blind channel-acquisition (BCA) method. Meanwhile, a soft information iterative process is used to improve the system performance. Finally, the convergence, bit error rate, and real-time performance of iterative detection can be further improved effectively by using improved exchange methods of extrinsic information and the stopping criterion. The analysis and simulation results show that the proposed algorithm achieves a good blind equalization performance and low complexity.

## 1. Introduction

Driven by the explosive surge of Internet of Things (IoT) services for sixth-generation (6G) mobile communications systems, different new 6G use cases have been proposed and are under intensive research discussion recently, such as IoT industry automation, maritime machine-type communication networks, and other applications [1,2]. As one of the key technologies to achieve the vision of the Internet of Everything, UAVs have been widely used to perform diversified tasks [3,4,5] due to their low cost and flexible deployment.

There has been a recent surge of studies on the use of UAVs for IoT communication [6,7,8], such as data collection [9,10] and mobile edge computing [11]. However, with the dramatic increase in the number of connected machines, the number of IoT devices deployed worldwide is expected to grow to 75.4 billion by 2025 [12]. There is a growing demand for low complexity and high power efficiency in UAV-aided IoT communication due to the limited payload of the devices.

Continuous phase modulation (CPM) is suitable for power- and bandwidth-limited systems because of its good spectral efficiency and its higher power efficiency relative to linear modulations with comparable spectral efficiency. Moreover, the constant envelope property of CPM allows the nonlinear power amplifier (PA) to be operated at a high efficiency, which further increases the power efficiency of the system [13]. For battery-powered IoT nodes and UAVs, energy efficiency and cost are key factors because these devices are difficult to recharge or recycle once depleted. Therefore, CPM is one of the preferred modulation schemes in UAV-aided IoT communications systems due to its favorable low power consumption, which can greatly increase the life of terminal devices.

However, CPM transmission over multipath fading channels is a challenging task due to the high computational complexity in the receiver. If the design of the waveform is poor, it will reduce the overall power of the communications systems, even offsetting the increased power efficiency achieved by the PA. Therefore, we focus our attention on the receiver design at the physical layer for CPM over frequency-selective channels employing low data rates and short bursty transmissions, which is a fundamental tool to implement UAV-aided IoT. In general, the main contributions of this paper include the following:We combed the literature related to CPM and summarized it in the Table 1.To meet the demands of low data rates and short-burst transmission scenarios of the UAV-aided IoT system, a short burst structure of CPM is designed in this paper, and a link-level simulation platform of the communications system is established on this basis.A low complexity approach for soft-input soft-output (SISO) blind equalization is proposed to achieve a fast and accurate blind equalizer in the UAV-aided IoT system. The first step utilizes the soft-output Lazy Viterbi algorithm instead of the Viterbi algorithm to perform the expectation step and obtain a low complexity expectation–maximization Lazy Viterbi algorithm (EMLVA), while the second step applies the BCA method to establish a set of initializers, denoted as the BCA initializers, which achieves a high global convergence probability.The blind turbo equalization for short-burst CPM is proposed based on the new SISO blind equalization with iterative detection, where the blind equalizer and decoder exchange extrinsic information in the form of log-likelihood ratios (LLRs). To further improve the convergence of iteration and reduce the average iteration number, the decision-aided (HDA) algorithm based on weighted extrinsic information exchange is proposed.The blind turbo equalization based on EMLVA is proposed and evaluated on a link-level simulation platform. Simulation results show that EMLVA can obtain a good trade-off between complexity and BER performance. When the HDA with weighted extrinsic information is applied, the convergence of iterative detection and real-time performance can be further improved.

The rest of this paper is organized as follows. Section 2 provides related work on the channel estimation and equalization of CPM transmission over frequency-selective channels. In Section 3, the burst structure, communications system, and channel model are designed and formulated for typical UAV-aided IoT communications scenarios. The new low-complexity blind equalization based on EMLVA, as well as the turbo scheme are introduced and described in Section 4. The performances of the proposed turbo blind equalizer are evaluated and analyzed based on the link-level simulation platform in Section 5. In Section 6, we discuss the application prospects of reflecting intelligent surfaces (RISs) in UAV-aided IoT communications systems. Finally, the paper is concluded in Section 7.

## 2. Related Work

In practical communications, the signals are transmitted over the fading channel and the channel response is unknown. In [14], a generalized pilot symbol-aided demodulation method based on the idea of inserting data-dependent symbols periodically was proposed for CPM in a flat Rayleigh fading channel. An optimal front-end filter was developed based on the mean-squared error (MSE) in the channel estimation process. Then, the channel estimates generated by the interpolation filter, together with the received signal, are input into a coherent CPM demodulator using the Viterbi algorithm. In frequency-flat fast-fading channels, Ref. [15] provided a data-aided channel estimation algorithm with local B-splines, and the results showed that there exists a minimum sampling interval proportional to the normalized fading rate for pilot insertion. However, when short bursts are considered, the data-aided channel estimation method can significantly increase the overhead-to-payload ratio. Similarly, low-complexity frequency-domain equalization for CPM [27,28,29] requires the addition of a cyclic prefix or unique words, which can also increase the overhead-to-payload ratio for short bursts.

As an alternative, blind channel equalization can recover the signal directly, without a training sequence. The author in [16] applied the Tong–Xu–Kailath algorithm to CPM by extracting the second-order statistics of the signal for channel estimation. The eigenvector method was used to identify the channel from a fourth-order cross-cumulant matrix under the GSM channel in [17], combined with turbo estimation. However, when applied to a low number of symbols, the statistical moments did not provide accurate channel estimation. In [18,30], the author developed a nonlinear signal model for GMSK rather than the conventional finite impulse response model. The information symbols were obtained by Bayesian inference based on Markov chain Monte Carlo (MCMC) with implicit channel estimation.

CPM and the multipath channel can construct a joint trellis, which can be represented by a finite state machine (FSM). Therefore, a forward adaptive SISO (FA-SISO) [19], which considers the channel correlation in only one direction, was proposed for MSK, which replaces the unknown channel by the least-mean-squared error for each hypothesis branch symbol. Then, due to the high complexity of FA-SISO, the author in [20] proposed various reduced-state A-SISO (RS-A-SISO) algorithms for complexity reduction at the same time. The thresholds of the RS-A-SISO algorithms were obtained by the density evolution technique in [21]. Another structure is the forward/backward adaptive algorithm. An exact expression for the soft metrics was derived when the unknown parameter was modeled as a Gauss–Markov process in [22], which can be estimated iteratively by the Kalman filter. The author in [23] employed the concept of bidirectional estimation in [22] and derived a generalized a posteriori probability of soft branch metrics.

The FSM can also be described by a hidden Markov model (HMM), and the Baum–Welch (BW)/EM algorithm allows for great likelihood estimation of the unknown parameters in the HMM. The batch-BW (BBW) algorithm, as well as two variants were proposed by Carles [24] for time-invariant channels. One is to split the received signal into several sub-blocks, producing different channel estimates in each, called the segmented batch-BW (SBBW) algorithm. However, the variant needs to avoid over-fragmentation because of a poor estimate from fewer data. An alternative algorithm called time-dependent BW (TDBW) was derived by introducing some linear constraints emerging from a linear FIR hypothesis on the channel. The author in [25] proposed an improved Baum–Welch algorithm to directly estimate the channel parameters, avoiding over-parameterization in the estimation problem. In [26], an algorithm for joint channel estimation and equalization by applying the Viterbi algorithm within an EM iteration was introduced, which was used to implement the E-step. However, the major drawbacks among the works cited above are relatively poor convergence with an inappropriate initializer and high complexity.

CPM serves as one of the preferred modulation schemes for the transmission of low data rates in the IoT uplink, suffering from the high complexity and poor convergence of the channel estimation at the receiver. Therefore, the paper proposes a low-complexity blind equalization algorithm for short-burst CPM signals based on the HMM. The proposed blind equalizer significantly outperforms the traditional one in complexity, while keeping a similar BER performance, which helps the device achieve online real-time detection. In general, as the spectrum resources are limited and the number of connected devices is increasing day by day, CPM is a promising modulation scheme, which is suitable for battery-powered devices and is expected to play an important role in the physical layer design of UAV-aided IoT communications.

## 3. System Model and Problem Description

### 3.1. Communications System Model

The communications system with blind turbo equalization is represented in Figure 1. It can be divided into three parts: transmitter, channel model, and blind turbo equalization:

(1) Transmitter: A sequence of message bits $m=(m_{0},m_{1},m_{2},\ldots ,m_{K})$ is encoded with the channel encoder into codeword $b=(b_{0},b_{1},b_{2},\ldots ,b_{V})$ and interleaved as $u=(u_{0},u_{1},u_{2},\ldots ,u_{V})$. Then, the block codewords are input to the M-ary CPM modulation module, which consists of the continuous-phase encoder (CPE) and memoryless modulator (MM). The baseband representation of CPM is written as
(1)$$s\left(t,\alpha_{n}\right)=\sqrt{\frac{E_{s}}{T}}\exp\left(j\phi \left(t,\alpha_{n}\right)\right),t\in \left[0,nT\right].$$
where $E_{s}$ and *T* represent the symbol energy and symbol duration, respectively. $\alpha_{n}$ is the CPM symbol sequence, containing n+1 symbols. The information-bearing phase is defined as
(2)$$\phi \left(t,\alpha_{n}\right)=2\pi h\sum_{i=0}^{n}\alpha_{i}q\left(t-iT\right).$$
*h* is the modulation index, defined as the quotient of two relatively prime integers. q(t) is the phase pulse obtained by the integral of the phase pulse g(t). *L* is the length of the frequency pulse, where L=1 is the full-response CPM and L>1 is the partial response.

(2) Channel: The CPM transmitted over the channel is affected by the fading channel and Gaussian white noise; the continuous representation of the received signal is
(3)r(t)=s(t,α)c(t)+w(t).

In digital processing, it is necessary to use a discrete-time representation. The sampling time is noted as $T_{s}=T/\varepsilon$, where ε is the number of sampling numbers per symbol. Therefore, the discrete-time representation of CPM, the received signal, and the noise can be defined as:(4)$$s_{n}={\left[s_{0},s_{1},s_{2}\ldots ,s_{n\varepsilon -1}\right]}^{T},$$
(5)$$r_{n}={\left[r_{0},r_{1},r_{2}\ldots ,r_{n\varepsilon -1}\right]}^{T},$$
(6)$$w_{n}={\left[w_{0},w_{1},w_{2}\ldots ,w_{n\varepsilon -1}\right]}^{T}.$$

The CPM signal for the *i*th path is defined as
(7)$$s_{n-i}={\left[0_{i},s_{0},w_{1},w_{2}\ldots ,s_{n\varepsilon -i-1}\right]}^{T}.$$
where $0_{i}$ is a row vector containing *i* zero elements and discrete channel length $N_{c}+1$. We define
(8)$$S_{n}\left(\alpha \right)={\left[s_{n},s_{n-1},\ldots ,s_{n-N_{c}}\right]}^{T}.$$
where $C^{n\times l}$ represents an *n*by*l* complex matrix. The channel response is $c={\left[c_{0},c_{1},\ldots ,c_{N_{c}}\right]}^{T}$, and then, the discrete-time representation of the received signal in the form of a matrix can be written as
(9)$$r_{n}=S_{n}\left(\alpha_{n}\right)c+w_{n}.$$

(3) Channel estimation and equalization: The CPM and multipath channel can build a joint trellis, which can be regarded as an FSM. According to Bayesian theory, the posterior probability is
(10)$$p\left(\alpha ,c|r\right)=\frac{p(r|\alpha ,c)p(\alpha )p(c)}{p(r)}.$$

Since the receiver does not have a priori information about the channel, the probability density function p(c) can be regarded as a constant. The received signal is known at the receiver, and as a result, p(r) is also a constant. Therefore, we can write p(α,c|r)∼p(r|α,c)p(α), where
(11)$$p\left(r|\alpha ,c\right)=\frac{1}{{\left(\pi \sigma_{w}^{2}\right)}^{\kappa n}}\exp\left(-\frac{1}{\sigma_{w}^{2}}||r-{S\left(\alpha \right)c||}^{2}\right).$$

With the perfect channel state information (CSI), maximum likelihood (ML) detection requires maximizing Equation (Equation 11). However, in practice, the channel response is unknown to the receiver and channel estimation is required.

The FSM can be described by a hidden Markov model because the input symbols are independent of each other. Therefore, the channel estimation and equalization for CPM can be transferred to the three classical problems of the hidden Markov model (HMM) to solve, where channel estimation can be regarded as the parameter learning problem of the HMM and channel equalization can be regarded as the decoding problem of the HMM.

(4) Turbo equalization: The receiver can be used in the form of turbo equalization because of the SISO Viterbi-like algorithm. The iterative exchange of information is repeated iteratively between the blind equalizer and the decoder for the same set of the received signal. The received signal r and the soft information $L_{D}\left(b_{t}\right)$ output from the decoder (initially set to 0) are input into the SISO blind equalizer, and then, the blind equalizer outputs the extrinsic LLR on the symbols $L_{E}\left(\alpha_{k}|r\right)$, which is fed to the decoder after interleaving. The decoder provides an estimation of the information bits $\hat{m}$ using soft decisions and computes an extrinsic LLR on the coded bits $L_{D}\left(b_{t}\right)$.
(12)$$L_{D}\left(b_{t}\right)=ln\left(\frac{b_{t}=1|L_{E}\left(b_{t}|r\right)}{b_{t}=0|L_{E}\left(b_{t}|r\right)}\right)-L_{E}\left(b_{t}|r\right).$$

### 3.2. Design of Burst Structure

The definition of the short in short bursts varies from a few tens to hundreds of bits in different literature [31]. The literature [32] designed a short convolutional code for machine-type communications, where the information word length was 64 bits and the codeword length was 128 bits. In [31], the authors considered a scenario with only a few tens of bits over the duration of a burst. The number of transmitted CPM symbols in a single burst is was symbols with a burst duration of 1 ms and a very high-frequency baud rate in [33]. The burst structure is shown in Figure 2, with the training sequence using only a small number of bits to quickly and reliably lockthe receiver, rather than for channel estimation in the conventional one. It is particularly important to note that the BER results in the simulation are a function of the average (information) bit-energy-to-noise-power density spectral ratio $E_{b}/N_{0}$. The following relationship holds
(13)$$\frac{E_{s}}{N_{0}}=\frac{E_{b}}{N_{0}}\cdot log_{2}\left(M\right)\cdot R_{c}\cdot \eta_{tr}.$$
where $E_{s}$ is the average symbol energy, M is the modulation base number of CPM, and $R_{c}$ is the code rate. A data-aided approach is used as a benchmark scheme in the subsequent simulations, with training bits for channel estimation and $\eta_{tr}=N_{tr}/N_{all}$ representing the power efficiency loss caused by the training bits.

### 3.3. Channel Model

For short bursts, a time-invariant channel was assumed throughout a burst when the UAV flies at medium and low speeds, and the channel in any two different bursts is independent. A typical communications scenario is shown in Figure 3. Considering the effects of topography, atmosphere, and other factors, the propagation shows the multipath fading problem. In a ground-to-air scenario for UAV-aided IoT communications, the UAV adopts the hover-and-fly method for data collection via the IoT uplink. In the communication between the UAV and the ground nodes, the multipath channel model can be described by a Ricean fading channel with the probability distribution of the Rician distribution:(14)$$f\left(r\right)=\left\{\begin{array}{ccc} \frac{r}{\sigma^{2}} & exp\left(-\frac{r^{2}+\alpha_{LOS}^{2}}{2\sigma^{2}}\right)I_{0}\left(\frac{r\alpha_{LOS}^{2}}{\sigma^{2}}\right), & 0\leq r<\infty \\ 0, & & r<0 \end{array}\right.$$
where *r* is the envelope of the fading signal, $\alpha_{LOS}$ is the amplitude of the LOS component, $\sigma^{2}$ represents the energy of the NLOS component, and $I_{0}$ is a modified Bessel function of the first kind with order zero. Rician factor $K_{Ricean}$ is defined as the power ratio of the main signal to the multipath components, where $K_{Ricean}=\alpha_{LOS}^{2}/\sigma^{2}$. In the IoT-aided UAV communications scenario, this paper refers to [34] for the air–ground (AG) channel at 5060 MHz (C baud), which corresponds to a Ricean factor of 29 dB. Then, the two-ray Rician channel model, including the LOS and NOLS components, can be specified as
(15)$$c\left(\tau ,t\right)=\alpha_{LOS}\delta \left(\tau -\tau_{1}\left(t\right)\right)+\alpha_{2}e^{-j\phi (t)}\delta \left(\tau -\tau_{2}\left(t\right)\right).$$

## 4. EMLVA for Blind Channel Equalization

### 4.1. EM Algorithm

The EM algorithm is divided into two steps, one of which is the expectation step (E-step) and the other is the maximization step (M step). For a blind equalization problem, the observations r and transmitted symbols α can be written as a new data vector z=(r,α). The EM algorithm then consists of the following iteration.

E-step: Compute the mean log-likelihood function:(16)$$Q\left(\theta |\theta^{\left(m\right)}\right)=E_{a}\left[lnf_{r_{n}}\left(r_{n}|\alpha ,\theta \right)|r_{n},\theta^{\left(m\right)}\right].$$
where f(·) represents a conditional probability density.

M step: Compute new estimates of parameters:(17)$$\theta^{\left(m+1\right)}=argmaxQ\left(\theta |\theta^{\left(m\right)}\right).$$

### 4.2. VA and Its Variants

The Viterbi algorithm (VA) was proposed in 1967 for maximum likelihood (ML) detection of convolutional codes, searching for the ML paths by dynamic programming. At each symbol duration, the path metrics of all paths entering the current state are compared by the VA, and the path with the largest metric is selected for that state, called the survivor path. Finally, the VA outputs the information bits corresponding to the survivor path, which is named the ML path. Thus, the computational complexity of the VA is $O(M^{L}N)$, where *M* is the number of states of the finite-state machine, *L* is the memory length, and *N* is the length of the sequence. When the number of states in the trellis is too large, searching on the original full trellis for ML paths requires a large number of computational resources. It is not always necessary to traverse the full trellis when only the ML path is searched, especially in the case of a high signal-to-noise ratio.

Before the VA, sequential decoding algorithms are used in decoding the convolutional code, which are essentially greedy algorithms. They only search for branches that are possible to become the ML path, which has low complexity, but cannot guarantee finding the ML path. Therefore, a variant of the VA algorithm, the Lazy VA algorithm [35,36], was developed. A priority queue (PQ) data structure is introduced to ensure finding ML paths. Compared to the standard VA, the Lazy VA is far better than the Viterbi algorithm in complexity and no worse than the VA in the worst case. For the ease of writing, the VA and Lazy VA are denoted as Viterbi-like algorithms in this paper, and the Lazy Viterbi algorithm is summarized in Algorithm 1.
**Algorithm 1** Lazy Viterbi algorithm [36].1:The trellis is set to empty, and the PQ contains the initial node $n_{s}$ with initialized cumulative metric $acc(n_{s})=0$.2:Pop the top node $n_{1}$ of PQ.3:**if**$n_{1}$ is the same as some node of the trellis **then**4:   Discard $n_{1}$5:**else if**$n_{1}$ with the smallest metric in the PQ **then**6:   Output as the current node7:**else if**$n_{1}$ is not the last node **then**8:   Insert its successors into PQ, and return to Step 29:**else**10:  Trace back the ML path and output hard/soft decision bits11:**end if**

### 4.3. The EMVA/EMLVA Blind Equalizer

With these definitions in mind, the Lazy VA in Algorithm 1 is used to implement the E-step for the EM algorithm. The resulting algorithm is denoted as the expectation–maximization Lazy VA (EMLVA) soft blind equalizer, and it can be found in Algorithm 2, which can be implemented as follows:
**Algorithm 2** The EMVA/EMLVA blind equalizer.1:**Input**: the received signal r2:Set the maximum estimated number of iterations *S*3:Initialization: $\theta^{\left(0\right)}=\left(c^{\left(0\right)},{N_{0}}^{\left(0\right)}\right)$4:**for**s=1 to *S* **do**5:   With the current initializer to run Viterbi or Lazy Viterbi in Algorithm 1 ${\left(a_{q}^{\left(s\right)},\Lambda_{q}^{\left(s\right)}\right)}_{q=1}^{Q}=Viterbi/LazyViterbi\left(r,\theta^{\left(s\right)}\right)$6:   Calculate the probability of each path by (18)7:   Compute the expectation of the logarithmic likelihood function by (19)8:   Maximize the expectation and compute the estimated parameters by (21) and (22)9:**  end for**10:**Output**: the soft information corresponding to the ML path

(1) Set the initializer $c^{\left(0\right)}=\left(c^{\left(0\right)}\left(0\right),c^{\left(0\right)}\left(1\right),...,c^{\left(D\right)}\left(0\right)\right)$ and $N_{0}^{\left(0\right)}$, the maximum number of inner iterations S, which is the iterations of EMLVA starting with zero.

(2) A set of Q survivor paths is obtained by using the Viterbi-like algorithm with current initializer $\theta^{\left(s\right)}=\left(c^{\left(s\right)},N_{0}^{\left(s\right)}\right)$. The probability of the *q*th survivor path can be computed:(18)$$P_{q}^{\left(s\right)}=Pr\left(a_{q}^{\left(s\right)}|r,\theta^{\left(s\right)}\right)=\frac{1}{z^{\left(s\right)}}exp\left(\eta^{\left(s\right)}-\frac{\Lambda_{q}^{\left(s\right)}}{{N_{0}}^{\left(s\right)}}\right)$$
where $\Lambda_{q}^{\left(s\right)}$ is the path metric for the *q*th survivor path, $\eta^{\left(s\right)}$ is a constant to prevent these metrics from becoming too large, which is usually selected as the minimum path metric, and the normalizing factor $z^{\left(s\right)}$ satisfies $\sum_{q}P_{q}^{\left(s\right)}=1$.

(3) Obtain the complete data expected log-likelihood function:(19)$$L\left(\theta |\theta^{\left(s\right)}\right)=E_{a}\left(LLR|r,\theta^{\left(s\right)}\right)\approx \sum_{q=1}^{Q}\frac{P_{q}^{\left(s\right)}||r-{S(\alpha )c||}^{2}}{N_{0}}-ln\left(N_{0}\right)+d.$$
where *d* is a constant independent of the estimated parameter θ.

(4) Maximize the likelihood function for $\theta =(c,N_{0})$:(20)$$\theta^{\left(s+1\right)}=argmaxL\left(\theta |\theta^{\left(s\right)}\right).$$
where
(21)$$c^{\left(s+1\right)}={\left(\sum_{q=1}P_{q}^{\left(s\right)}S^{H}\left(a_{q}\right)S\left(a_{q}\right)\right)}^{-1}\cdot \left(\sum_{q=1}P_{q}^{\left(s\right)}S^{H}\left(a_{q}\right)\right)r.$$
(22)$${N_{0}}^{\left(s+1\right)}=\frac{1}{L}\sum_{q=1}P_{q}^{\left(s\right)}\Lambda_{q}^{\left(s\right)}.$$

(5) With the values $\left(c^{\left(S\right)},{N_{0}}^{\left(S\right)}\right)$ obtained after the EMV algorithm, make soft decisions as to which sequence of channel symbols or output soft information in input into the decoder.

Similarly, the standard VA used to implement the E-step for the EM algorithm is referred to as EMVA. A block diagram of the EMLVA/EMVA equalizer is included in Figure 4, where the grey boxes represent the initializer.

### 4.4. BCA Method and Convergence Criterion

Since the EM algorithm is sensitive to the initializer and the channel is different in any two different bursts, a single fixed initializer has a poor convergence, which falls into an initial value trap, making it difficult to track the channel. Therefore, a set of initializers based on the BCA method [37] is used in this paper. For complex channel response, global convergence is ensured with a high probability if the initializer contains only one nonzero unit real-valued tap and one nonzero unit purely imaginary tap, located at the appropriate locations. Therefore, for a complex channel response of length *l*, there are ${\left[\left(2l+1\right)\right]}^{2}$ initializers for a set, and the number is reduced to (2l+1) for a real-valued channel. The set of initializers is denoted as the BCA initializer, and the other is called a single initializer.

When the set of initializers has been traversed, a set of estimated parameters is obtained, and then, the optimal estimated parameters need to be selected. For the *k*th initializer, the EMV equalizer converges to $\hat{\theta_{k}}={\left(\hat{c},\hat{N_{0}}\right)}_{k}$, and the evaluation of the likelihood function can be expressed as
(23)$$L\left(\theta_{k}\right)=-Lln{\hat{N}}_{0,k}-\frac{{\hat{\eta }}_{k}}{{\hat{N}}_{0,k}}+ln{\tilde{L}}_{k}.$$
where
(24)$${\tilde{L}}_{k}=\sum_{q=1}^{Q}exp\left[\frac{{\hat{\eta }}_{k}-{\hat{d}}_{k,q}}{{\hat{N}}_{0,k}}\right],$$
(25)$${\hat{d}}_{k,q}=||r-S\left(a_{q}\right){\hat{c}}_{k}{\left|\right|}^{2},$$
(26)$${\hat{\eta }}_{k}=min{\hat{d}}_{k,q}.$$

The best estimated parameters can be selected by maximizing $L(\theta_{k})$, i.e.,
(27)$$\hat{\theta }=\theta_{\hat{k}},\hat{k}=argmaxL\left(\theta_{k}\right).$$

### 4.5. The Turbo EMLVA Blind Equalizer and Positive Feedback

Figure 4b shows that the detector can further improve performance through turbo equalization in Section 3.1, which can effectively improve the bit error rate performance and convergence of the algorithm. A turbo equalization EMV algorithm based on the BCA initializer is presented in Algorithm 3. To distinguish the iteration of turbo equalization from the iteration of the EM algorithm, the turbo equalization between CPM-SISO and Decoder-SISO is denoted as the outer iteration, and the iteration of the EM algorithm is called the inner iteration. The algorithm is as follows:
**Algorithm 3** The EMLVA blind turbo equalizer (T-EMLVA).1:**Input**: the received signal r2:Set the maximum outer iteration equalization times *T*3:**for**t=0 to *T* **do**4: With the current initialization to $\theta^{\left(0\right)}=\left(c^{\left(0\right)},{N_{0}}^{\left(0\right)}\right)$, run Algorithm 25:  Compute an estimate to the extrinsic LLRs $L_{E}\left(b_{t}|(r\right))$. Feed $L_{E}\left(b_{t}|(r\right))$ to the channel decoder6:   From the channel decoder per-bit soft-output, recompute LLRs for each symbol $L_{D}\left(\alpha_{k}\right)$7:**end for**8:**Output**: Output the soft decision output detection results

(1) A set of initializers $\theta_{k}^{\left(0\right)}={\left(c^{\left(0\right)},N_{0}^{\left(0\right)}\right)}_{k},k=1,2,\ldots ,K$, and select the *k*th initial value $\theta^{\left(0\right)}={\left(c^{\left(0\right)},N_{0}^{\left(0\right)}\right)}_{k}$.

(2) Set the maximum iteration number of turbo equalization $T_{max}$, with the initial outer iteration T=0; set the maximum iteration number of iterations of the EM algorithm $S_{max}$ with the initial inner iteration s=0.

(3) With the current initializer θ, the received signal *r* and the a priori information $L_{D}\left(u\right)$ returned by the convolutional code (initially 0), run Algorithm 2 to obtain the Q survivor paths ${\left\{a_{q}\right\}}_{q=1}^{Q}$, the path metric ${\left\{\Lambda_{q}\right\}}_{q=1}^{Q}$, and the soft information ${\left\{L_{E}^{q}\left(\alpha \right)\right\}}_{q=1}^{Q}$ of each path.

(4) After traversing all the initializers, the optimal estimated parameter is selected by the convergence criterion in Equation (Equation 27) for the blind equalization, and the corresponding soft symbol information $L_{E}\left(\alpha \right)$ is demapped and deinterleaved to obtain the soft information $L_{E}\left(b_{t}\right)$ of the information bits, which is input into the channel decoder as a priori information.

(5) When $T<T_{max}$, an extrinsic LLR on the coded bits $L_{D}\left(b\right)$ is mapped and interleaved again and delivered back to the CPM blind equalization as the updated a priori probability. Steps 3 to 5 are repeated for a given maximum number of iterations $T_{max}$.

For turbo equalization, the inner iteration using different Viterbi algorithms is denoted as T-EMVA and T-EMLVA, respectively. It should be noted that under the condition of a short burst, the coded CPM system exhibits positive feedback in the process of outer iteration and convergence to a suboptimal solution by directly exchanging extrinsic information. Therefore, extrinsic information exchange methods play an important role in the convergence of the outer iteration, and the performance of equalization can be further improved by superior extrinsic information exchange methods.

### 4.6. Complexity Analysis

The computational complexity of the EMV based on the BCA initializer mainly comes from the Viterbi-like algorithm of the E-step. Only considering this part, the complexity of the EMVA is $O(\left[2{\left(D+1\right)}^{2}SpM^{L+L_{c}-1}N\right])$, and the complexity after adding the outer iteration of turbo equalization is $O({\left[2\left(D+1\right)\right]}^{2}SpM^{L+L_{c}-1}N+TSM^{L+L_{c}-1}N)$, where *D* is the channel memory length, $L_{c}$ is the lengthin terms of the symbol time, *L* is the memory length of the CPM frequency pulse, *M* is the base number of CPM, *S* is the maximum number of inner iterations, *T* is the maximum number of external iterations, and *N* is the sequence length.

To reduce the calculation amount, the EMLVA can be implemented by applying the Lazy VA in the E-step. In the best case, the complexity can be reduced to $O({\left[2\left(D+1\right)\right]}^{2}SN)$, and T-EMLVA has complexity of $O({\left[2\left(D+1\right)\right]}^{2}SN+TSN)$. After adding the iteration stopping criterion, the number of outer iterations can be further reduced and the decoding delay can be effectively reduced. The average number of iterations of turbo equalization after adding the stopping criterion can be denoted as $T_{ave}$, and the computational amount is $O({\left[2\left(D+1\right)\right]}^{2}SN+T_{ave}SN)$. A detailed comparison of the complexity of the T-EMVA and the T-EMLVA is included in Table 2.

## 5. Experimental Evaluation

### 5.1. Experimental Setup

For the evaluation of the proposed blind turbo equalization, the parameters are summarized in Table 3 and the burst structure is shown in Figure 2. GMSK with the modulation index h = 1/2 was taken as an example for the experiment and simulation. The channel was modeled as a two-ray Ricean channel model, which set the maximum delay of 7μs; to better simulate it in the hilly/mountainous scenarios for UAV-aided IoT systems, the Ricean factor was chosen as 29 dB. The sample time with a baud rate of 150 kHz was 3.3μs, and then, the symbol duration was T=6.6μs. For the maximum delay 7μs, the channel length at the receiver was D+1=⌊7/3.3⌋+1=3 taps, while the channel length in terms of symbol duration was $L_{c}=\left\lfloor D/\varepsilon \right\rfloor +1=2$; the sign ⌊⌋ represents rounding down.

### 5.2. Simulation Results

#### 5.2.1. System Parameter Optimization

A serially concatenated CPM (SCCPM) setup with iterative decoding was considered in the numerical simulations. In view of the insufficiency of interleaving for the short burst, the optimized parameters of the convolutional code (CC) and interleaver need to be selected.

The number of states and the free distance of a CC can affect the interleaving gain and the performance of the iterative detection. In general, a CC with a larger number of states has a larger free distance and can further provide more interleaving gain. However, this only occurs when the interleaving length is large enough and the signal-to-noise ratio is above the convergence threshold. When the frame length is short, a CC with a large number of states exhibits poor convergence, leading to performance degradation. Therefore, Figure 5a compares the BER performance of five types of CC, i.e., (5,7), (17,15), (23,35), (53,75), and (171,131) after 0 outer iterations and 10 outer iterations. The result shows that (5,7) CC is the optimal channel code for the SCCPM system, taking into account the burst length, signal-to-noise ratio, and implementation complexity.

The interleaver was used to randomize the order of the code bits before transmission. To select the appropriate interleaver to enhance the performance gain, Figure 5b compares the performance of six types of interleavers: matrix interleaver, random interleaver, S-random interleaver, general interleaver, QPP interleaver, and WCDMA interleaver. The analysis of the BER curves for 0 and 10 outer iterations showed that the S-random interleaver can provide the best coding gain and performance. Therefore, the S-random interleaver was used for subsequent simulations.

#### 5.2.2. Performance Comparison of EMLVA, EMVA, and the Method Based on the Training Sequence

Figure 6a compares the BER performance of the data-aided coherent detection, EMVA, and EMLVA proposed in this paper with different inner and outer iterations, as well as the SOVA with perfect channel state information (CSI). The BER curves with perfect channel knowledge represent the maximum attainable performance, while the curves for the data-aided method (called Tran.Bits.) represent the baseline to beat, which extracts the channel by using maximum likelihood estimation with training sequence lengths of 4, 6, or 8 bits, respectively. The performance of the data-aided method was lower than that of the blind equalization EMLVA proposed in this paper after 10 outer iterations, indicating that the data-aided approach has difficulty obtaining an accurate channel with few training bits. Compared to the traditional EMVA, the performance loss of the EMLVA was small, which can be ignored and approximates the curve with perfect channel knowledge after 10 outer iterations.

As seen in Figure 6b, the BCA-initializer-based EMLVA can ensure good convergence of the blind equalization with a fixed inner iteration of three. More outer iterations did not improve the performance significantly after about five iterations, which can be viewed as convergence. On the contrary, when a single initializer was used, the BER curves decreased slowly as $E_{b}/N_{0}$, and there was not a considerable performance improvement in the iterative process. The performance loss can be explained by the poor convergence to track changing channels in short bursts.

Figure 7 analyses the normalized-mean-squared error (NMSE) of the EMVA and EMLVA based on the BCA initializer, as well as the NMSE of the EMLVA using a single initializer and the NMSE based on the training bits method. It can be seen that the proposed EMLVA equalizer with the BCA initializer significantly outperformed the EMVA in complexity while keeping a similar estimation accuracy as the EMVA. Moreover, the NMSE of the EMLVA based on a single initializer had poor convergence, and there was no decreasing trend at high $E_{b}/N_{0}$. At the same time, the NMSE curves of the data-aided method with few training bits exhibited poor convergence and estimation accuracy, although there was a decreasing trend.

#### 5.2.3. Improved Exchange Methods of Extrinsic Information and Stopping Criterion

Direct exchange is an original method, which can be further improved by weighted exchange. In weighted exchange, the extrinsic information $L_{E}$ is sent to the weighted function module before being delivered to the other SISO module. The weighted function is expressed as:(28)$$W\left(L_{E}\right)=\alpha L_{E}*exp(-\beta |L_{E}\left|\right).$$

In order to evaluate the effect of different weighting coefficients, the simulation was implemented with values of α∈[0.8,1.0] and β∈[0.001,0.01]. The BER performance with various weighted coefficient combinations of α and β is shown in Figure 8a–c.

Figure 8d shows the BER as a function of $E_{b}/N_{0}$ for different extrinsic information, including weighted extrinsic information, average extrinsic information, and direct methods, with or without the HDA stopping criterion. The results show that there was little loss of performance of the algorithm with HAD criterion and the combination of α=0.9 and β=0.01 had the best BER performance.

#### 5.2.4. The Effect of Channel Length Overestimation on the Performance

Finally, the effect of channel length overestimation on the performance was investigated. Up till this point, the channel length was assumed known. However, in practice, this is not the case. The BER performance of the EMLVA with different initializer schemes is analyzed in Figure 9a when the assumed channel length is correct (${\hat{N}}_{c}=0$) and when it is overestimated (${\hat{N}}_{c}=1,2,3$). The results illustrate the apparent robustness of the EMLVA with the BCA initializer compared to the EMLVA with a single initializer.

Figure 9b shows the number of expansions performed by the EMLVA and EMVA as a function of $E_{b}/N_{0}$, where the channel overestimation length is expressed in symbol time. It is easy to see that the number of states traversed by the EMVA increased exponentially with the overestimation length, while the number of states searched by the EMLVA remained constant for high $E_{b}/N_{0}$. Even in the event of severe channel length overestimation, the complexity of the EMLVA can be decoupled from the memory length above a certain $E_{b}/N_{0}$. Therefore, Figure 9b shows the superiority of the EMLVA proposed in this paper in terms of complexity, which can meet the computational requirements of low complexity for IoT nodes.

## 6. Discussion

In this paper, we focused our attention on the physical layer design of a UAV-aided IoT communications system. In recent years, an emerging and revolutionizing technology, RIS, can significantly improve communication performance by smartly reconfiguring the wireless propagation environment [38].

In the non-LOS scenarios of UAV-aided IoT, the RIS can be applied to maximize the received power of the user to keep the connection. In the presence of eavesdroppers, the reflected signal by the RIS can be tuned to cancel out the signal from the sensors node at the eavesdropper by smartly adjusting the reflection coefficients [39]. Apart from this, when the RIS is applied in the UAV-aided IoT communications system, the structure of the transmitter and receiver can be simplified, further meeting the requirements of low power and cost [40]. Predictably, the application of the RIS to future UAV-aided IoT communications systems will fundamentally change their architecture and significantly improve their performance.

## 7. Conclusions

In this paper, the EMLVA based on the HMM was proposed as an efficient approach to blind equalization of short CPM bursts in a UAV-aided IoT communications scenario. The proposed method significantly outperformed the HMM blind equalization based on the standard Viterbi algorithm in complexity, while keeping a similar BER performance as the conventional one. An initializer was developed based on the BCA method, which ensured good convergence of channel estimation, called the BCA initializer. The EMLVA equalization can be further improved by using the decoder output to propose the turbo EMLVA equalization. Considering the convergence and relatively large decoding delay in outer turbo iterations, weighted extrinsic information with the HDA stopping criterion is proposed for iterative detection. The simulation results showed that the proposed blind turbo equalization achieved an excellent trade-off between complexity and performance, verifying its advantages and practical values against other conventional methods.

## Figures and Tables

**Figure 1 sensors-22-06508-f001:**
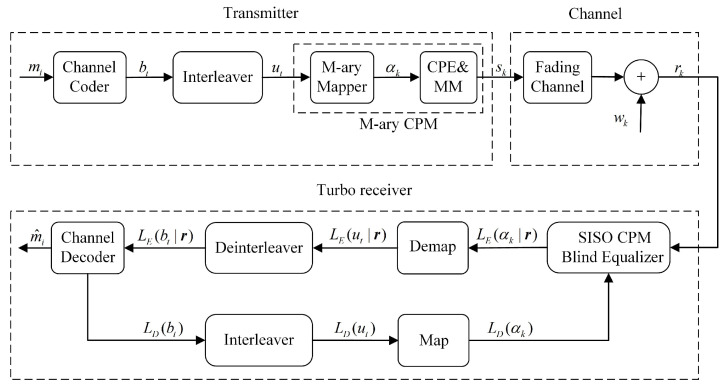
Communications system model, including the transmitter, channel, and turbo receiver.

**Figure 2 sensors-22-06508-f002:**
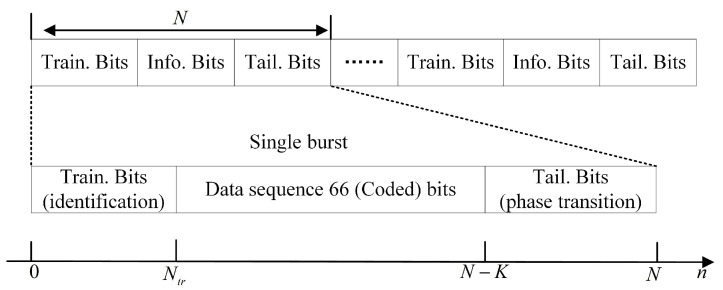
Block diagram of the transmitted CPM burst signal. Labels Train. and Tail. refer to the training and transition symbols, respectively.

**Figure 3 sensors-22-06508-f003:**
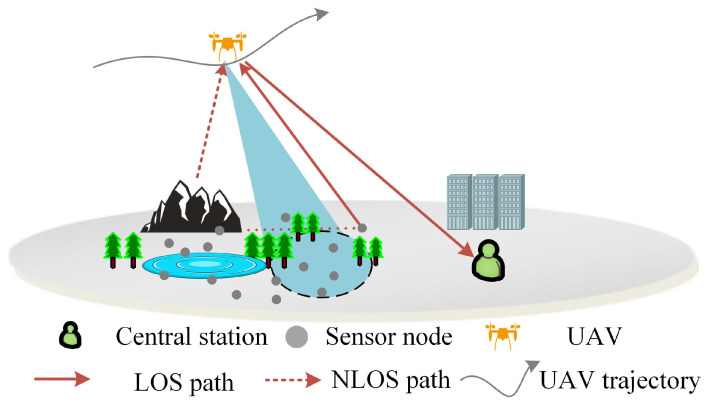
Typical UAV-aided IoT communications scenario with G2A cases.

**Figure 4 sensors-22-06508-f004:**
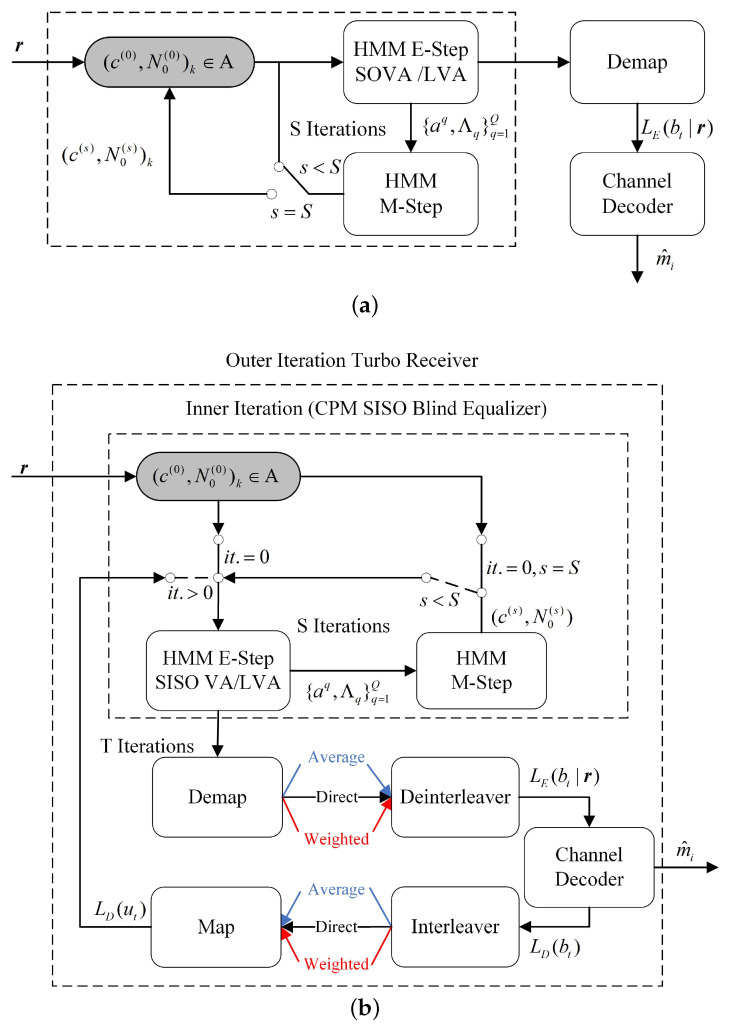
In (**a**), EMLVA blind equalizer block diagram. In (**b**), turbo EMLVA blind equalizer block diagram.

**Figure 5 sensors-22-06508-f005:**
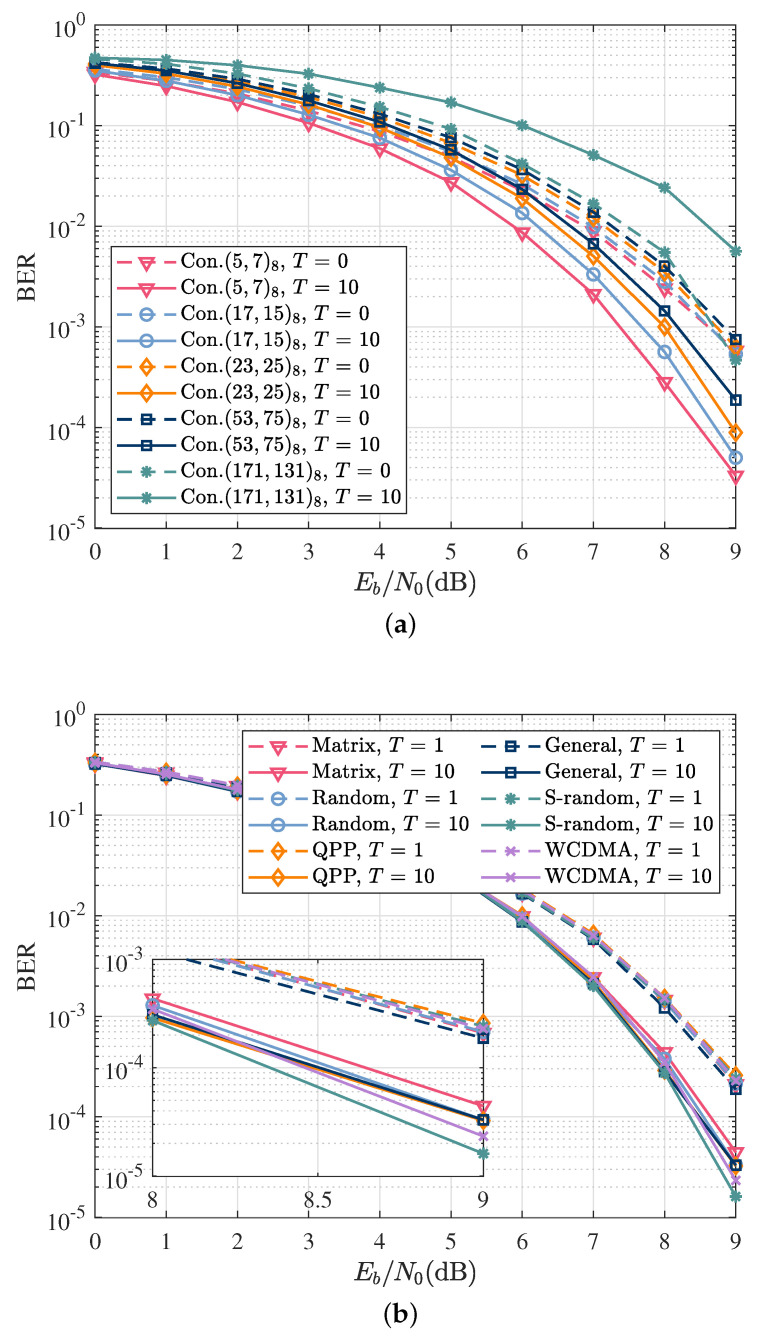
The performance of different convolutional codes and interleavers with perfect channel knowledge. The dashed and solid curves refer to the performance of the optimal coherent detector after the first iteration (T=1) and the tenth iteration (T=10), respectively. (**a**) BER as a function of $E_{b}/N_{0}$ for different convolutional codes; (**b**) BER as a function of $E_{b}/N_{0}$ for different interleavers.

**Figure 6 sensors-22-06508-f006:**
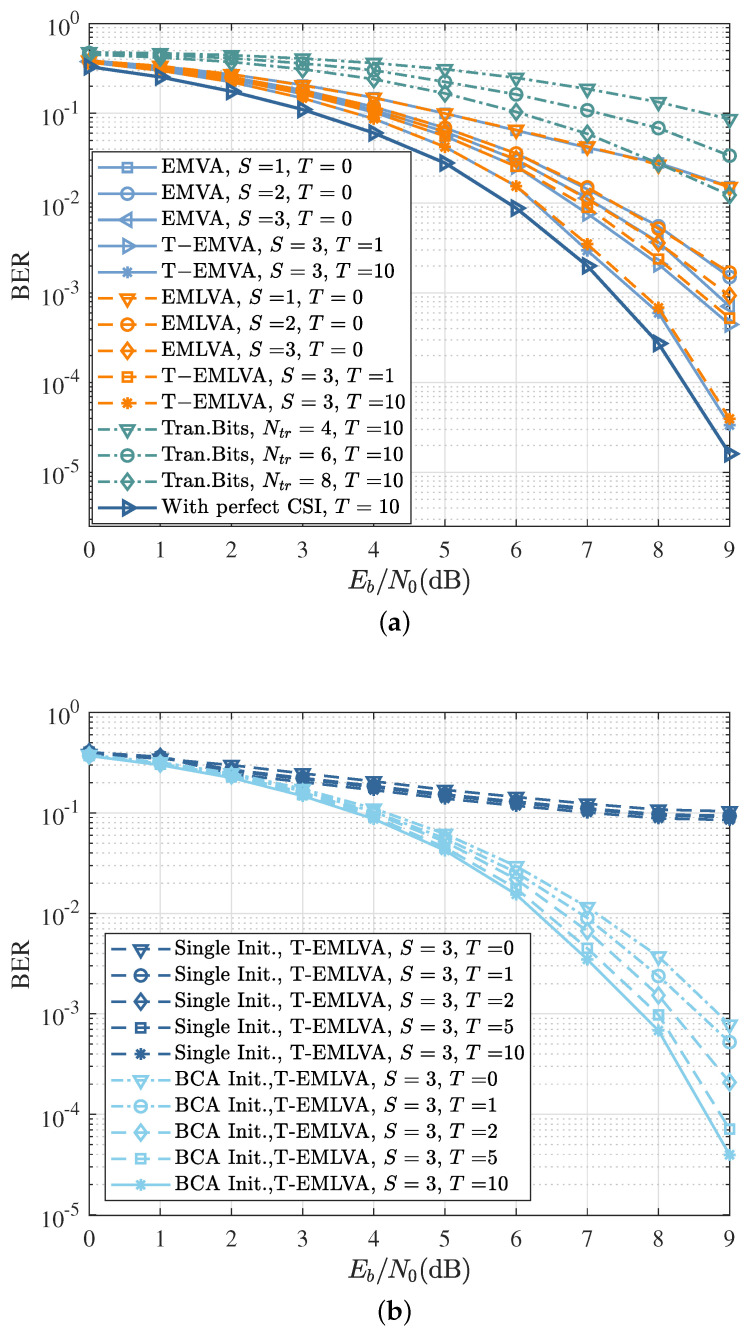
The performance of the EMLVA and T-EMLVA. (**a**) BER as a function of $E_{b}/N_{0}$ for the EMLVA and T-EMLVA with the BCA initializer. The “With perfect CSI” and “Tran. Bits” curves refer to the performance of the optimal coherent detector and training-based channel estimation, respectively; (**b**) BER as a function of $E_{b}/N_{0}$ for the EMLVA and T-EMLVA with different initializers.

**Figure 7 sensors-22-06508-f007:**
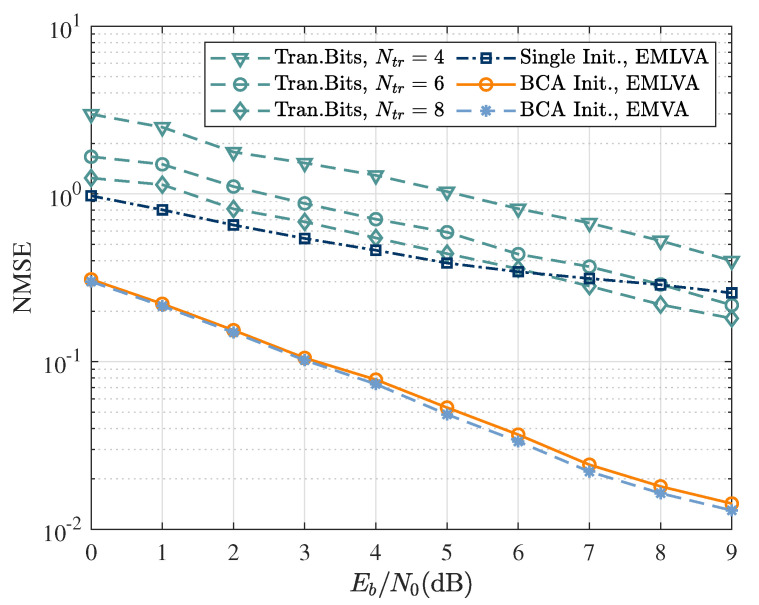
NMSE as a function of $E_{b}/N_{0}$ for training-based channel estimation. EMVA with single fixed initializer; EMLVA and EMVA with BCA initializer.

**Figure 8 sensors-22-06508-f008:**
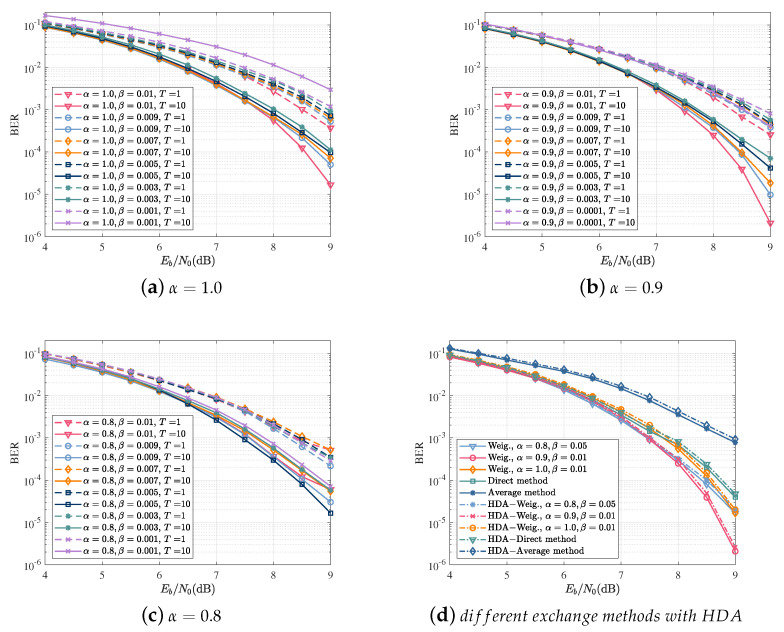
The performance of different exchange methods of extrinsic information with or without a stopping criterion. (**a**) BER as a function of $E_{b}/N_{0}$ for α=1.0; (**b**) BER as a function of $E_{b}/N_{0}$ for α=0.9; (**c**) BER as a function of $E_{b}/N_{0}$ for α=0.8; (**d**) BER as a function of $E_{b}/N_{0}$ for different exchange methods of extrinsic information with the HDA stopping criterion.

**Figure 9 sensors-22-06508-f009:**
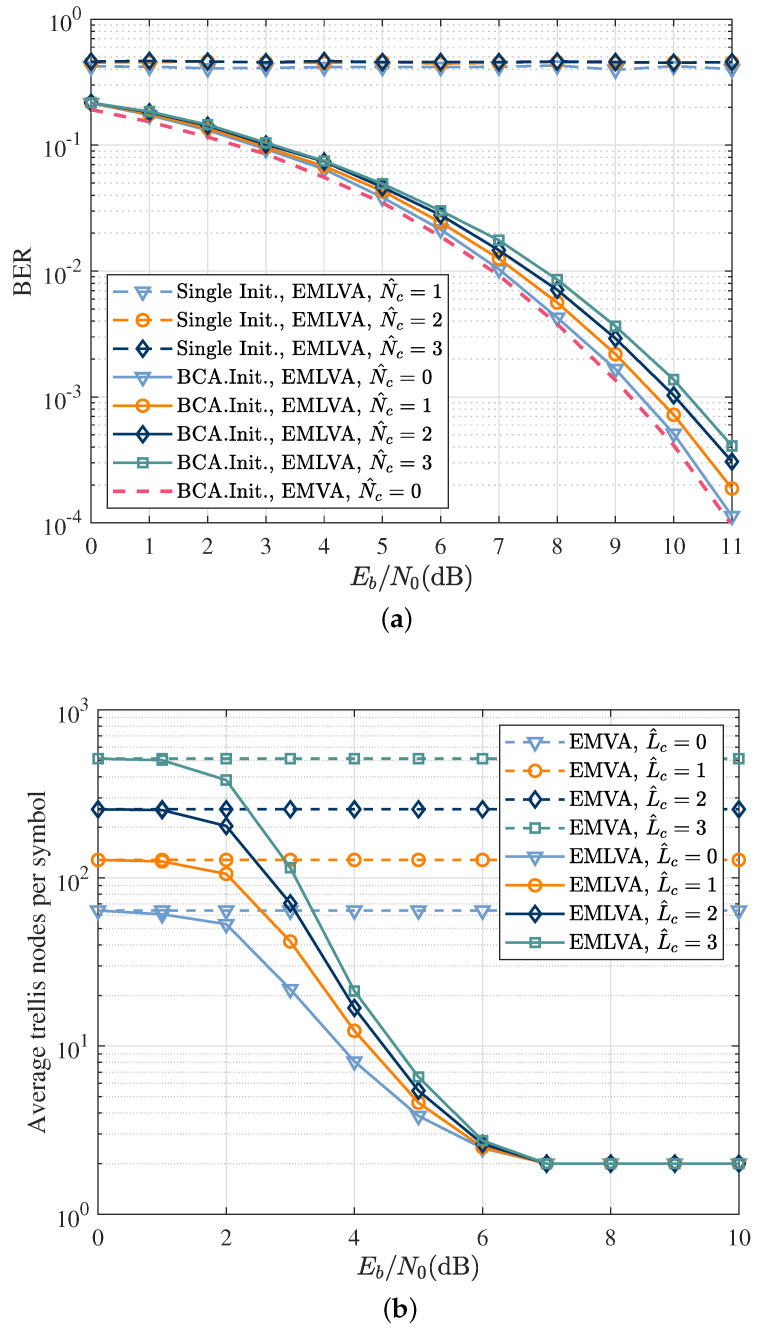
The performance of the EMLVA when the channel length is overestimated. (**a**) BER as a function of $E_{b}/N_{0}$ for the blind equalizer in a Ricean fading channel when the channel length at the receiver is known (${\hat{N}}_{c}=0$) and when it is overestimated (${\hat{N}}_{c}=1,2,3$). BER as a function of $E_{b}/N_{0}$ when the channel length is overestimated. (**b**) Average number of expanded trellis nodes per information symbol as a function of $E_{b}/N_{0}$ when the channel length at the receiver is known (${\hat{L}}_{c}=0$) and when it is overestimated in terms of the symbol time (${\hat{L}}_{c}=1,2,3$). Average trellis nodes per symbol as a function of $E_{b}/N_{0}$ when the channel length is overestimated.

**Table 1 sensors-22-06508-t001:** Summary of the related work.

Classification	Ref.	Contribution/Methodology
Data-aided	[14]	A generalized pilot symbol-aided demodulation method is proposed in a flat fading channel. The optimal filters for channel estimation are also presented.
	[15]	The estimate of the channel is realized by local B-splines.
Statistics	[16]	The second-order statistics of the signal for channel estimation is extracted for CPM by TXK.
	[17]	A fourth-order cross-cumulant matrix is extracted by the eigenvector method.
MCMC	[18]	A nonlinear signal model for GMSK and information symbols with implicit channel estimation by MCMC are developed.
Adaptive equalization	[19]	A forward adaptive SISO that considers the channel correlation in only one direction is proposed for MSK.
	[20]	A variety of the reduced state FA SISO is proposed.
	[21]	The thresholds of the RS-A-SISO algorithms are obtained by the density evolution technique.
	[22]	Derivation of the forward/backward adaptive algorithm.
	[23]	Derivation of the generalized forward/backward adaptive algorithm.
HMM	[24]	The BBW algorithm, as well as two variants, are proposed for CPM.
	[25]	A stochastic ML blind channel estimation is developed, and an approximate Cramér–Rao bound for CPM is derived.
	[26]	The Viterbi algorithm is applied within the EM algorithm.
FDE	[27]	The single-carrier frequency-domain equalization is used in the CPM signal for the first time.
	[28]	Laurent decomposition is used to realize traditional equalization (linear and decision feedback) and turbo equalization in the frequency domain.
	[29]	More iterative gain without matrix inversion.

**Table 2 sensors-22-06508-t002:** Complexity comparison of the algorithms.

Algorithm	Standard Equalization	Turbo Equalization
EMVA	$O({\left[2\left(D+1\right)\right]}^{2}SpM^{L+L_{c}-1}N)$	$O({\left[2\left(D+1\right)\right]}^{2}SM^{L+L_{c}-1}N+TSpM^{L+L_{c}-1}N)$
EMLVA	$O({\left[2\left(D+1\right)\right]}^{2}SN)$	$O({\left[2\left(D+1\right)\right]}^{2}SN+TSN)$
EMLVA with HDA	$O({\left[2\left(D+1\right)\right]}^{2}SN)$	$O({\left[2\left(D+1\right)\right]}^{2}SN+T_{ave}SN)$

**Table 3 sensors-22-06508-t003:** Simulation parameters.

Parameters	Value	Remarks
Frequency pulse	Gaussian pulse	L=3
Modulation order (M)	2	-
Modulation index (h)	1/2	h=p/q
Training bits	4,6,8	ML estimation
Coded data bits	66	-
Baud rate	150 kHz	-
Code rate	1/2	convolutional code
Samples/symbol	2	ε
Multipath channel	two-ray Rician channel	-
Rician factor	$K_{Ricean}=29\;\mathrm{dB}$	hilly/mountainous scenarios
Maximum delay spread	7μs	-
Inner iteration (S)	3	$S_{max}$
Outer iteration (T)	10	$T_{max}$

## Data Availability

Not applicable.

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
