# Peer review of "Blind Turbo Equalization of Short CPM Bursts for UAV-Aided Internet of Things"

_sensors, 2022, doi:10.3390/s22176508_

Round 1

Reviewer 1 Report

In this paper, a blind equalization of short continuous phase modulated (CPM) burst is proposed for UAV-aided IoT. The following are the comments.

1.The variable {\epsilon} on line 131 of page 4 lacks of definition.

2. The variable {\bf{S}_{n}(\alpha)} in formula (8) is a matrix. Therefore, the dimensions of this matrix should be specified explicitly.

3. It is pointed in page 8 that the computational complexity is O(M^{L}N). When the number of paths L is large, the complexity of the algorithm is very high. In the simulation, with L=2, the complexity is acceptable. Is the algorithm practical if L is increased, for example using the channel in Table I of Ref. [7]?

4. In the simulation, the performance of the proposed algorithm should be compared with the frequency domain processing method, such as the method in References [7]-[9].

Reviewer 2 Report

In this paper, the EMLVA based on HMM is proposed as an efficient approach to blind equalization of short CPM bursts in a UAV-aided IoT communication scenario. The proposed method significantly outperforms HMM blind equalization based on the standard Viterbi algorithm in complexity while keeping a similar BER performance as the conventional one. I have the following comments:

1) The related work needs further improvement by citing more recent works. Currently, authors only cite a few papers, and the contribution of this work is doubted. Some work are:

a) Feng, W., Wang, J., Chen, Y., Wang, X., Ge, N. and Lu, J., 2018. UAV-aided MIMO communications for 5G Internet of Things. IEEE Internet of Things Journal6(2), pp.1731-1740.; b) Guo, H. and Liu, J., 2019. UAV-enhanced intelligent offloading for Internet of Things at the edge. IEEE Transactions on Industrial Informatics16(4), pp.2737-2746.; c) Li, K., Ni, W., Tovar, E. and Guizani, M., 2020. Joint flight cruise control and data collection in UAV-aided Internet of Things: An onboard deep reinforcement learning approach. IEEE Internet of Things Journal8(12), pp.9787-9799.; d) Zheng, X., Zhang, J. and Pan, G., 2022. On Secrecy Analysis of Underlay Cognitive UAV-Aided NOMA Systems with TAS/MRC. IEEE Internet of Things Journal.; e) Haider, S.K., Jiang, A., Almogren, A., Rehman, A.U., Ahmed, A., Khan, W.U. and Hamam, H., 2021. Energy efficient UAV flight path model for cluster head selection in next-generation wireless sensor networks. Sensors21(24), p.8445. f) Lu, X., Yang, W., Yan, S., Li, Z. and Ng, D.W.K., 2021. Covertness and Timeliness of Data Collection in UAV-Aided Wireless-Powered IoT. IEEE Internet of Things Journal. g) AlJubayrin, S., Al-Wesabi, F.N., Alsolai, H., Duhayyim, M.A., Nour, M.K., Khan, W.U., Mahmood, A., Rabie, K. and Shongwe, T., 2022. Energy Efficient Transmission Design for NOMA Backscatter-Aided UAV Networks with Imperfect CSI. Drones6(8), p.190.

There are so many other works. Please add all related works.

2) When the author adds all related papers to the literature review, it would be great if they include a table for comparison between different works. This table will also include the contribution of this work.

3) In the first paragraph of the introduction, the author say that "Driven by the explosive surge of Internet of Things (IoT) services for sixth-generation (6G) mobile communication systems, different 6G new use cases have been proposed and under intensive research discussions recently, such as IoT industry automation, maritime machine-type communication network, and other applications." however, they did not cite any related work for supporting it. I would recommend citing the paper "Efficient power allocation for NOMA-enabled IoT networks for 6G era, Physical Communication, 2020."

4) The explanation of the Algorithm is not clear. It is suggested to revise the discussion for the reader's understanding. In addition, the other algorithms can be checked for better understanding. 

5) The simulation results are not compared with any literature work. Is it possible to compare the performance of the proposed framework directly with recent works in the literature?

6) The authors only study the BER of the system even though there exist other important performance metrics, such as energy efficiency, trajectory, spectral efficiency, and security of the system. 

7) Reflecting intelligent surfaces is one of the emerging technology for secure and energy-efficient UAV communication, i.e., "Opportunities for physical layer security in UAV communication enhanced with intelligent reflective surfaces, arXiv preprint arXiv:2203.16907" read and report this study and explain how it can work in your model?

8) Extensive proofreading is suggested for typos and grammar errors.

Round 2

Reviewer 2 Report

Thank you so much for addressing my comments.